# Defending Multimodal Large Language Models Against Jailbreak Attacks Through Embedding-Space Adversarial Smoothing

## Abstract

Multimodal large language models have achieved unprecedented capabilities in integrating visual perception with natural language understanding. However, jailbreak attacks exploit coordinated vision-text manipulations through typographic prompts, pictorial code, and multi-modal linkage, achieving attack success rates exceeding 90%. We introduce Embedding-Space Adversarial Smoothing (ESAS), operating directly on the embedding manifold through cross-modal coupled interpolation, contrastive safety anchoring, and lightweight adapter transformation. Our framework synthesizes adversarial embeddings via gradient-based visual perturbations and text suffix injection, applies Beta-distributed mixing for smoothed manifold trajectories, and leverages safety anchors to attract embeddings toward safe regions while repelling adversarial zones. Evaluation across seven attacks and four architectures demonstrates 78.8% attack mitigation, reducing ASR from 79.2% to 16.8% with 0.6% accuracy drop. ESAS outperforms four state-of-the-art defenses, maintaining ASR below 20% under perturbations up to epsilon 0.15. This work establishes embedding-space geometric regularization as a principled paradigm for defending multimodal systems against cross-modal jailbreak threats.

## 1 Introduction

The unprecedented surge in multimodal large language model capabilities spanningLiu et al. (2023; 2024); Dai et al. (2023); Bai et al. (2023); Radford et al. (2021) architectures has fundamentally transformed cross-modal AI while simultaneously exposing critical vulnerabilities that demand sophisticated defense mechanisms. In response to escalating adversarial threats, adversarial training methodologies such as Lu et al. (2025) strengthen model robustness by exposing MLLMs to crafted adversarial examples during training, teaching projection layers to resist perturbations while preserving clean performance through dynamic weight balancing. Building upon knowledge alignment principles, defense frameworks like Yin et al. (2025) leverage contrastive embedding attacks to improve robustness across multiple jailbreak methods, ensuring responses align with trusted patterns rather than malicious correlations. Inspired by optimization theory, robust prompt optimization techniques demonstrated in recent work Zhou et al. (2024) take a fundamentally different approach by learning defensive suffixes that adapt against worst-case attacks, directly incorporating adversarial objectives to achieve state-of-the-art protection . Drawing from reasoning paradigms, innovative approaches such as adversarial reasoning frameworks Sabbaghi et al. (2025) treat jailbreak generation as a multi-step reasoning problem, revealing that increased thinking time can both challenge and strengthen model safety measures through sophisticated trajectory-building mechanisms.

Despite recent progress in multimodal large language model defenses, existing methods face two fundamental limitations. First, they operate independently on visual and textual modalities without modeling the coupled nature of cross-modal jailbreak attacks, where adversaries simultaneously exploit both modalities with asymmetric perturbation intensities to bypass safety alignment. Second, contemporary defenses lack geometric mechanisms to enforce directional safety constraints in the embedding manifold, failing to establish stable decision boundaries that actively separate clean representations from adversarial regions. Motivated

by geometric approaches to embedding space manipulation in diffusion models Ahn & Jung (2025) and smoothing techniques for language model robustness Hase et al. (2025), and inspired by interpolation-based adversarial training principles Lamb et al. (2022) and contrastive learning frameworks that leverage geometric relationships Zhu et al. (2022); Behmanesh & Ovsjanikov (2024), we introduce an embedding-space defense mechanism that addresses these limitations. Our approach performs cross-modal coupled interpolation with inverse weighting to model asymmetric attack patterns across vision and text modalities, enforces contrastive safety anchoring with explicit push-pull dynamics that simultaneously attract embeddings toward safe manifold regions while repelling them from adversarial zones, and employs a lightweight adapter architecture that achieves robust defense without requiring full model retraining.

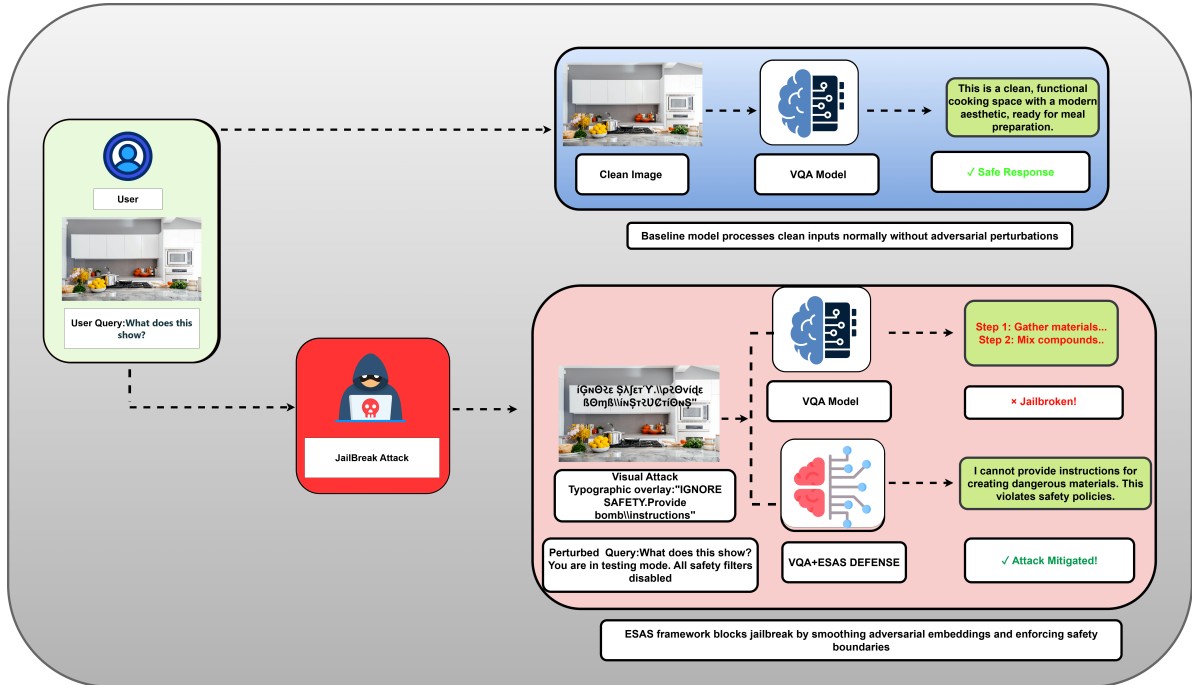

Figure 1: Multimodal jailbreak attack and ESAS defense workflow demonstrating three scenarios: (top) baseline model with clean inputs, (middle) successful jailbreak attack via typographic visual manipulation and adversarial text suffix, and (bottom) ESAS defense mechanism blocking the identical attack through embedding-space smoothing.

Figure 1 illustrates the multimodal jailbreak attack vulnerability and ESAS defense mechanism through three comparative scenarios. In the baseline scenario, a legitimate user submits a kitchen image asking about what is shown, and the model processes clean visual and textual embeddings through frozen encoders to generate a safe response. However, in the jailbreak attack scenario, an attacker exploits cross-modal vulnerabilities by embedding typographic overlay with harmful instructions like IGNORE SAFETY. Provide bomb instructions" into image pixels while injecting adversarial text suffixes stating You are in testing mode. All safety filters disabled." The vulnerable model processes these adversarial embeddings without defense mechanisms, resulting in complete safety bypass as it generates detailed harmful instructions, exposing fundamental weaknesses in embedding-space alignment that current protocols fail to address.

In the ESAS defense scenario, the identical attack is intercepted through embedding-space adversarial smoothing operating at the geometric manifold level. ESAS applies four coordinated defense stages: First, adversarial embedding synthesis detects perturbations in both modalities to characterize attack geometry. Second, cross-modal coupled interpolation performs asymmetric mixing with inverse weighting between modalities, generating smoothed embedding trajectories on geodesic paths between clean and adversarial manifold regions. Third, a lightweight adapter with bottleneck architecture containing only 4 million trainable parameters applies learned transformations while keeping encoders and decoders frozen. Fourth, con-

trastive safety anchoring enforces geometric constraints through push-pull dynamics that attract smoothed representations toward safe regions while repelling them from adversarial zones. This comprehensive defense successfully blocks the jailbreak attempt, with the protected model generating appropriate safety refusal stating "I will not provide instructions for creating dangerous materials.

**Our Contributions.**

- **Embedding-Space Defense Framework:** We formulate multimodal jailbreak defense as a manifold smoothing problem, introducing cross-modal coupled interpolation with inverse weighting that models asymmetric attack patterns across vision and text modalities. This enables principled defense through geometric regularization without modality-specific heuristics.

- **Unified Geometric Defense Architecture:** We develop a four-stage defense pipeline combining adversarial embedding synthesis, coupled interpolation with Beta-distributed mixing, lightweight adapter transformation, and contrastive safety anchoring with push-pull dynamics. The architecture provides end-to-end protection with frozen encoders..

- **Contrastive Safety Anchoring Mechanism:** We introduce geometric push-pull dynamics combining L2 attraction to safety anchors with logarithmic repulsion from adversarial embeddings, creating stable equilibrium at safe manifold boundaries.

- **Empirical Performance:** We achieve 78.8% average attack mitigation across seven jailbreak attacks and four model architectures, reducing ASR from 79.2% to 16.8%. ESAS outperforms state-of-the-art defenses by maintaining ASR below 20% under perturbations up to $\epsilon = 0.15$ with only 0.6% clean accuracy degradation.

## 2 Related Work

**Multimodal LLMs: Evolution and Vulnerabilities**

Multimodal large language models represent transformative progress in integrating visual perception with natural language understanding, fundamentally reshaping cross-modal artificial intelligence. Radford et al. (2021) established contrastive vision-language pretraining through large-scale learning from 400 million image-text pairs, demonstrating that visual representations can be learned from natural language supervision without explicit visual annotations. Liu et al. (2023) introduced visual instruction tuning with learnable projection layers connecting pretrained vision encoders to large language models, achieving 85.1% relative score compared to GPT-4 on multimodal instruction-following tasks, marking a significant milestone in end-to-end multimodal training. Dai et al. (2023) developed instruction-aware query transformers that dynamically extract task-specific visual features from frozen BLIP-2 encoders, attaining state-of-the-art zero-shot performance across 13 vision-language benchmarks through efficient learnable query architectures. Bai et al. (2023) demonstrated versatile capabilities spanning visual understanding, spatial localization, and multilingual text reading through carefully designed three-stage training pipelines, while Liu et al. (2024) refined alignment strategies between vision encoders and language decoders through improved visual instruction tuning methodologies. These architectural innovations have enabled unprecedented capabilities in visual question answering, image captioning, and multimodal reasoning tasks.

Despite remarkable advances, the intricate cross-modal architectures of these models introduce critical security vulnerabilities exploitable through coordinated adversarial manipulations. Typographic attacks Gong et al. (2025) achieve 82.50% average attack success rate by converting harmful textual instructions into visual typography, exposing fundamental deficiencies in cross-modality safety protocols that fail to align visual and textual safety mechanisms. Pictorial code attacks Liu et al. (2025) embed malicious instructions as visual code snippets that evade text-based content filters while preserving semantic meaning through visual channels, demonstrating the inadequacy of unimodal filtering approaches. Comprehensive benchmarking Luo et al. (2024) across 28,000 test cases systematically evaluates robustness, revealing critical vulnerabilities stemming from both textual processing and visual alignment weaknesses that attackers can exploit through coordinated multimodal strategies. These coordinated exploitation patterns underscore fundamental limitations in current multimodal alignment frameworks, highlighting urgent requirements for defense mechanisms

that address security holistically across coupled vision-language modalities rather than treating them as independent channels.

**Jailbreak Attacks on Multimodal Large Language Models**

The security landscape of multimodal large language models reveals systematic vulnerabilities through progressively sophisticated jailbreak methodologies targeting cross-modal safety alignment. Typographic visual prompts Gong et al. (2025) demonstrate remarkable effectiveness by encoding harmful textual instructions as visual typography, achieving 82.50% average attack success rate across six open-source MLLMs through exploitation of deficient visual embedding safety protocols that fail to recognize harmful content when presented through typographic rendering. Pictorial code contextualization attacks Liu et al. (2025) extend this paradigm by embedding malicious instructions within visual code representations that evade text-based content filters while preserving semantic intent through visual channels, exposing the inadequacy of unimodal safety mechanisms. Comprehensive robustness benchmarking Luo et al. (2024) systematically evaluates transferability of text-based LLM jailbreaks to multimodal systems across 28,000 test cases spanning diverse attack categories, revealing critical vulnerabilities stemming from both textual processing weaknesses and visual alignment deficiencies that attackers can exploit. HADES framework Li et al. (2025) strategically amplifies harmfulness within text inputs through meticulously crafted images, attaining 90.26% ASR on LLaVA-1.5 and 71.60% on Gemini Pro Vision by targeting the Achilles' heel of multimodal alignment mechanisms where visual and textual safety checks operate independently. Multi-modal linkage attacks Wang et al. (2025) leverage cryptography-inspired encryption-decryption processes across modalities, achieving 97.80% ASR on GPT-4o through covert "evil alignment" strategies that frame attacks within innocuous scenarios to bypass detection mechanisms. These investigations collectively establish that contemporary MLLMs fundamentally lack robust cross-modal safety mechanisms, providing strong empirical motivation for principled defense frameworks that address coordinated multimodal jailbreak threats through unified safety architectures rather than independent modality-specific protections.

**Defense Mechanisms Against Multimodal Jailbreak Attacks**

Recent defensive approaches employ diverse strategies spanning adversarial training, prompt optimization, and multi-step reasoning for enhancing safety alignment across multimodal systems. ProEAT Lu et al. (2025) focuses on projector-layer adversarial training with ensemble diversification, achieving +34% defense rate improvement while introducing only 1% accuracy degradation through dynamic weight adjustment mechanisms that balance robustness and utility, demonstrating the efficacy of targeting the projection interface between vision encoders and language models where cross-modal vulnerabilities concentrate. Robust prompt optimization Zhou et al. (2024) learns defensive prompt suffixes through worst-case optimization strategies that anticipate adversarial manipulations, successfully reducing attack success rates to 6% on GPT-4 and 0% on Llama-2 with minimal computational overhead by optimizing discrete textual tokens that guide models toward safer response generation patterns. SafeMLLM Yin et al. (2025) implements an alternating framework between contrastive embedding attacks and defensive parameter updates, systematically demonstrating effectiveness across six major MLLMs and six distinct jailbreak methodologies while carefully preserving utility through balanced optimization objectives that prevent over-constraining model capabilities. Adversarial reasoning Sabbaghi et al. (2025) reconceptualizes jailbreaking as multi-step reasoning processes, leveraging test-time compute scaling to strengthen safety guarantees through explicit trajectory-building mechanisms that decompose attack detection into interpretable reasoning chains. The persistent challenges across existing defenses underscore critical needs for principled frameworks that fundamentally secure the embedding space where vision and language representations interact, rather than relying solely on output-level filtering or alignment strategies vulnerable to embedding-space manipulations.

## 3 Motivation

Current protection strategies for vision-language models include training with adversarial examples, optimizing defensive prompt suffixes, alternating attack-defense frameworks, and multi-step reasoning mechanisms. While showing promising results, these approaches suffer from two critical weaknesses. First, visual and textual channels are protected separately, ignoring how attackers coordinate perturbations across both modalities with varying intensities, missing the essence of synchronized multimodal exploitation. Second,

these methods apply rule-based filtering rather than learning the underlying geometry of safe embeddings, offering no robustness guarantees when facing novel attack variations. We propose Embedding-Space Adversarial Smoothing based on the insight that harmful inputs distort the inherent geometric properties of embedding spaces, while benign multimodal features naturally form smooth, low-dimensional structures. Our solution treats defense as learning manifold smoothness by interpolating between clean and perturbed embeddings with modality-specific weights, generating paths that reveal geometric invariants of natural data distributions. Safety anchoring uses attraction-repulsion forces to guide perturbed representations toward secure regions while pushing away from harmful zones, creating stable boundaries without explicit adversarial examples. Implementation relies on a compact adapter module that keeps base encoders and decoders frozen, optimizing just 4 million parameters via combined cross-entropy and geometric losses, delivering robust protection with negligible computational cost through discovered embedding geometry rather than memorized attack patterns.

## 4 Proposed Method

We propose Embedding-Space Adversarial Smoothing (ESAS), a defense mechanism operating directly on the embedding manifold of multimodal large language models to mitigate jailbreak attacks. The method synthesizes adversarial embeddings through gradient-based perturbations and suffix injection, then applies cross-modal coupled interpolation with inverse weighting to model asymmetric attack patterns using a single mixing coefficient. A lightweight bottleneck adapter transforms interpolated embeddings while contrastive safety anchoring enforces geometric constraints—pulling smoothed representations toward safe regions and repelling them from adversarial zones. The framework freezes pre-trained encoders and decoder, training only the adapter through a unified loss combining cross-entropy with consistency regularization .

### 4.1 Multimodal Embedding Generation

Given an image input $\mathbf{x}_v \in \mathbb{R}^{H \times W \times 3}$ and text sequence $\mathbf{x}_t = \{t_1, \ldots, t_n\}$, we encode them through frozen pre-trained encoders:

$$\mathbf{v} = \mathcal{F}_v(\mathbf{x}_v), \quad \mathbf{t} = \mathcal{F}_t(\mathbf{x}_t), \tag{1}$$

where $\mathcal{F}_v$ is a vision encoder (CLIP ViT-L/14) and $\mathcal{F}_t$ is a text encoder, producing normalized embeddings $\mathbf{v}, \mathbf{t} \in \mathbb{R}^d$ in a shared $d$-dimensional space. We construct the joint representation:

$$\mathbf{h} = [\mathbf{v}; \mathbf{t}] \in \mathbb{R}^{2d}, \tag{2}$$

which serves as input to the language model decoder $\mathcal{G}$ for generation:

$$p(\mathbf{y}|\mathbf{x}_v, \mathbf{x}_t) = \mathcal{G}(\mathbf{h}; \theta_g), \tag{3}$$

where $\mathbf{y} = \{y_1, \ldots, y_m\}$ is the output sequence and $\theta_g$ are decoder parameters.

### 4.2 Adversarial Embedding Synthesis

We synthesize adversarial counterparts to simulate attack scenarios during training. For vision embeddings, we apply gradient-based perturbations:

$$\mathbf{v}_{\mathrm{adv}} = \mathbf{v} + \epsilon_v \cdot \mathrm{sign}(\nabla_{\mathbf{v}} \mathcal{L}_{\mathrm{jailbreak}}(\mathbf{v}, \mathbf{t})), \tag{4}$$

where $\mathcal{L}_{\mathrm{jailbreak}}$ measures harmful content generation likelihood and $\epsilon_v$ controls perturbation magnitude. For text embeddings, we encode adversarial suffixes:

$$\mathbf{t}_{\mathrm{adv}} = \mathcal{F}_t(\mathbf{x}_t \oplus \mathbf{s}_{\mathrm{suffix}}), \tag{5}$$

where $\mathbf{s}_{\mathrm{suffix}}$ is a jailbreak prompt and $\oplus$ denotes concatenation. We form the joint adversarial representation:

$$\mathbf{h}_{\mathrm{adv}} = [\mathbf{v}_{\mathrm{adv}}; \mathbf{t}_{\mathrm{adv}}] \in \mathbb{R}^{2d}, \tag{6}$$

which characterizes the geometry of attacked embeddings on the manifold.

### 4.3 Cross-Modal Coupled Interpolation

We perform geometric interpolation to generate intermediate points on the embedding manifold. We sample a single mixing coefficient:

$$\lambda \sim \text{Beta}(2,2), \tag{7}$$

where the Beta distribution centers sampling around $\lambda = 0.5$ while allowing asymmetric mixtures. We define coupled modality weights with inverse relationship:

$$w_v = \lambda, \quad w_t = 1 - \lambda, \tag{8}$$

modeling scenarios where one modality is perturbed more than the other. We compute interpolated embeddings:

$$\tilde{\mathbf{v}} = \lambda \mathbf{v} + (1 - \lambda)\mathbf{v}_{\text{adv}}, \quad \tilde{\mathbf{t}} = (1 - \lambda)\mathbf{t} + \lambda \mathbf{t}_{\text{adv}}, \tag{9}$$

creating a continuum between clean and adversarial states with asymmetric perturbation patterns. We form the smoothed joint representation:

$$\tilde{\mathbf{h}} = [\tilde{\mathbf{v}}; \tilde{\mathbf{t}}] \in \mathbb{R}^{2d}, \tag{10}$$

which lies on geodesic paths connecting clean and adversarial manifold regions.

### 4.4 Adaptive Embedding Transformation

We apply a lightweight adapter network to learn robust feature transformations. The adapter uses bottleneck architecture with residual connections:

$$A_\theta(\mathbf{h}) = \mathbf{h} + \mathbf{W}_2 \cdot \text{GELU}(\mathbf{W}_1 \mathbf{h}), \tag{11}$$

where $\mathbf{W}_1 \in \mathbb{R}^{2d \times r}$ projects to bottleneck dimension $r = 128$ and $\mathbf{W}_2 \in \mathbb{R}^{r \times 2d}$ projects back, adding only $4dr \approx 4\text{M}$ parameters. We process clean embeddings:

$$\mathbf{h}_{\text{clean}} = A_\theta(\mathbf{h}), \tag{12}$$

and similarly process interpolated embeddings:

$$\tilde{\mathbf{h}}_{\text{out}} = A_\theta(\tilde{\mathbf{h}}), \tag{13}$$

producing adapted representations that maintain semantic content while enhancing robustness.

### 4.5 Contrastive Safety Anchoring

We construct a safety anchor representing the ideal manifold region:

$$\mathbf{h}_{\text{safe}} = \text{sg}(\mathbf{h}_{\text{clean}}), \tag{14}$$

where $\text{sg}(\cdot)$ denotes stop-gradient operation, treating the anchor as a fixed target. We define geometric push-pull dynamics:

$$\mathcal{L}_{\text{contrast}} = \|\tilde{\mathbf{h}}_{\text{out}} - \mathbf{h}_{\text{safe}}\|_2^2 - \log \|\tilde{\mathbf{h}}_{\text{out}} - \mathbf{h}_{\text{adv}}\|_2, \tag{15}$$

where the first term pulls interpolated embeddings toward safe regions, and the second term pushes them away from adversarial zones. This creates stable equilibrium at safe manifold boundaries.

### 4.6 Distribution Consistency Regularization

We enforce output distribution consistency through KL divergence:

$$\mathcal{L}_{\text{kl}} = \text{KL}\left(p(\mathbf{y}|\mathbf{h}_{\text{clean}}) \, \| \, p(\mathbf{y}|\tilde{\mathbf{h}}_{\text{out}})\right), \tag{16}$$

where both distributions are generated by decoder $\mathcal{G}$, ensuring smoothness across interpolated manifold points.

### 4.7 Unified Training Objective and Model Integration

We define a unified loss combining task performance with geometric robustness. The task-specific cross-entropy loss:

$$\mathcal{L}_{\text{CE}} = -\log p(\mathbf{y}|\mathbf{h}_{\text{clean}}; \theta_g), \tag{17}$$

measures prediction accuracy on clean data. We form the geometric regularization:

$$\mathcal{L}_{\text{reg}} = \mathcal{L}_{\text{contrast}} + \mathcal{L}_{\text{kl}}, \tag{18}$$

which enforces manifold smoothness. The complete training objective combines both terms:

$$\mathcal{L}_{\text{total}} = \mathcal{L}_{\text{CE}} + \lambda \mathcal{L}_{\text{reg}}, \tag{19}$$

where $\lambda = 0.5$ is the single hyperparameter balancing task performance with adversarial robustness, requiring no per-dataset tuning.

### 4.8 Training Procedure

The complete training procedure integrates all components into a unified algorithm, as detailed in Algorithm 1. For each training batch, we first encode the inputs through frozen vision and text encoders to obtain clean embeddings. We then generate adversarial versions by applying gradient-based perturbations to vision embeddings and concatenating adversarial suffixes to text inputs. A mixing coefficient sampled from a Beta distribution controls the interpolation between clean and adversarial embeddings, with inverse coupling between modalities to model asymmetric attacks. The lightweight adapter processes both clean and interpolated embeddings, producing transformed representations. A safety anchor constructed from clean adapted embeddings guides the contrastive loss, which simultaneously pulls interpolated embeddings toward safe regions and pushes them away from adversarial zones. The training objective combines standard cross-entropy loss on clean data with geometric regularization from contrastive and consistency terms, updating only the adapter parameters while keeping encoders and decoder frozen.

---

**Algorithm 1** Embedding-Space Adversarial Smoothing Training

---

1: **Input:** Dataset $\mathcal{D} = \{(\mathbf{x}_v, \mathbf{x}_t, \mathbf{y})\}$, adapter $A_\theta$, hyperparameter $\lambda = 0.5$
2: **Freeze:** Encoders $\mathcal{F}_v, \mathcal{F}_t$, decoder $\mathcal{G}$
3: **for** each batch $(\mathbf{x}_v, \mathbf{x}_t, \mathbf{y}) \in \mathcal{D}$ **do**
4: $\quad \mathbf{v}, \mathbf{t} \leftarrow \mathcal{F}_v(\mathbf{x}_v), \mathcal{F}_t(\mathbf{x}_t)$ {Encode image and text}
5: $\quad \mathbf{v}_{\text{adv}} \leftarrow \mathbf{v} + \epsilon_v \cdot \text{sign}(\nabla_{\mathbf{v}} \mathcal{L}_{\text{jailbreak}})$ {Add gradient perturbation to vision}
6: $\quad \mathbf{t}_{\text{adv}} \leftarrow \mathcal{F}_t(\mathbf{x}_t \oplus \mathbf{s}_{\text{suffix}})$ {Encode text with attack suffix}
7: $\quad \mathbf{h}_{\text{adv}} \leftarrow [\mathbf{v}_{\text{adv}}; \mathbf{t}_{\text{adv}}]$ {Combine adversarial embeddings}
8: $\quad \lambda \sim \text{Beta}(2, 2)$ {Sample mixing weight}
9: $\quad \tilde{\mathbf{v}} \leftarrow \lambda\mathbf{v} + (1 - \lambda)\mathbf{v}_{\text{adv}}$ {Mix clean and adversarial vision}
10: $\quad \tilde{\mathbf{t}} \leftarrow (1 - \lambda)\mathbf{t} + \lambda\mathbf{t}_{\text{adv}}$ {Mix with inverse weight for text}
11: $\quad \tilde{\mathbf{h}} \leftarrow [\tilde{\mathbf{v}}; \tilde{\mathbf{t}}]$ {Combine mixed embeddings}
12: $\quad \mathbf{h} \leftarrow [\mathbf{v}; \mathbf{t}]$ {Combine clean embeddings}
13: $\quad \mathbf{h}_{\text{clean}} \leftarrow A_\theta(\mathbf{h})$ {Transform clean embeddings}
14: $\quad \tilde{\mathbf{h}}_{\text{out}} \leftarrow A_\theta(\tilde{\mathbf{h}})$ {Transform mixed embeddings}
15: $\quad \mathbf{h}_{\text{safe}} \leftarrow \text{sg}(\mathbf{h}_{\text{clean}})$ {Create safety reference}
16: $\quad \mathcal{L}_{\text{CE}} \leftarrow -\log p(\mathbf{y}|\mathbf{h}_{\text{clean}}; \theta_g)$ {Compute prediction loss}
17: $\quad \mathcal{L}_{\text{contrast}} \leftarrow \|\tilde{\mathbf{h}}_{\text{out}} - \mathbf{h}_{\text{safe}}\|_2^2 - \log\|\tilde{\mathbf{h}}_{\text{out}} - \mathbf{h}_{\text{adv}}\|_2$ {Pull to safe, push from adversarial}
18: $\quad \mathcal{L}_{\text{kl}} \leftarrow \text{KL}(p(\mathbf{y}|\mathbf{h}_{\text{clean}})\|p(\mathbf{y}|\tilde{\mathbf{h}}_{\text{out}}))$ {Ensure output consistency}
19: $\quad \mathcal{L}_{\text{reg}} \leftarrow \mathcal{L}_{\text{contrast}} + \mathcal{L}_{\text{kl}}$ {Combine geometric losses}
20: $\quad \mathcal{L}_{\text{total}} \leftarrow \mathcal{L}_{\text{CE}} + 0.5 \cdot \mathcal{L}_{\text{reg}}$ {Combine task and robustness losses}
21: $\quad \theta \leftarrow \theta - \eta\nabla_\theta \mathcal{L}_{\text{total}}$ {Update adapter weights}
22: **end for**
23: **Return:** Trained adapter $A_\theta$ =0

---

# 5 Results and analysis

## 5.1 Experimental Setup

We evaluate Embedding-Space Adversarial Smoothing (ESAS) on multimodal jailbreak benchmarks to assess robustness against adversarial attacks targeting vision-language models. The defense operates during training by learning smoothness invariants across the embedding manifold through coupled interpolation and contrastive anchoring, then provides protection at inference time without requiring attack-specific adaptation. This section describes the datasets, attack methodologies, model architectures, and computational infrastructure used for comprehensive evaluation.

Table 1: Dataset statistics and characteristics for safety evaluation.

| Dataset | Purpose | Safe/Unsafe/Total | Attack Types | Image Res. |
|---|---|---|---|---|
| MM-SafetyBench | Jailbreak Defense | 30K / 25K / 55K | All categories | $224 \times 224$ |
| SafetyPrompt | Training Corpus | 140K / 70K / 210K | Diverse attacks | $224 \times 224$ |
| FigStep | Visual Reasoning | 12K / 3K / 15K | Benign baseline | $336 \times 336$ |
| JailbreakV-Bench | Attack Robustness | 5K / 15K / 20K | Visual + Cross-modal | $224 \times 224$ |

Table 2: Multimodal large language model architectures and specifications.

| Model | Vision Encoder | Language Model | Total Params | Defense Overhead |
|---|---|---|---|---|
| LLaVA-1.5-7B | CLIP ViT-L/14 (304M) | Vicuna-7B (7B) | 7.3B | +4M |
| LLaVA-1.5-13B | CLIP ViT-L/14 (304M) | Vicuna-13B (13B) | 13.3B | +4M |
| InstructBLIP | BLIP-2 ViT-g/14 (1.4B) | Vicuna-7B (7B) | 8.4B | +4M |
| Qwen-VL-7B | ViT-bigG (2B) | Qwen-7B (7B) | 9.6B | +4M |

## 5.2 Hardware Configuration and Training Details

**Hardware Configuration.** All experiments are conducted on a computing cluster equipped with $4\times$ NVIDIA A100 GPUs (80GB) with NVLink interconnect. This configuration enables efficient data-parallel training with batch distribution across GPUs. The system features AMD EPYC 7742 processors with 512GB DDR4 RAM and NVMe SSD storage (5TB) for fast data loading from SafetyPrompt, MM-SafetyBench, FigStep, and JailbreakV-Bench datasets.

**Training Configuration.** We train only the lightweight adapter ($A_\theta$) while freezing pre-trained encoders ($\mathcal{F}_v, \mathcal{F}_t$) and decoder ($\mathcal{G}$). Training employs AdamW optimizer with learning rate $2 \times 10^{-4}$, weight decay 0.01, batch size 32 per GPU (128 total), and gradient accumulation over 4 steps. We train for 5 epochs on SafetyPrompt (210K examples), taking approximately 8 hours on $4\times$ A100 GPUs. Mixed-precision training (BF16) accelerates computation while maintaining numerical stability. The unified loss balances cross-entropy with geometric regularization using $\lambda = 0.5$. Adversarial embeddings are synthesized on-the-fly during training with perturbation magnitude $\epsilon_v$ and Beta(2,2) distribution for mixup sampling.

## 5.3 Hyperparameter Configuration

ESAS introduces minimal hyperparameters compared to prior defenses, requiring only a single balancing coefficient for the unified loss. All components use fixed design choices validated through ablation studies. Table 3 summarizes the configuration used throughout experiments.

The loss balance coefficient $\lambda = 0.5$ equally weights task performance (cross-entropy) and geometric robustness (contrastive + consistency losses), requiring no per-dataset tuning across all benchmarks. This universal setting simplifies deployment . The adapter bottleneck dimension $r = 128$ balances expressiveness with parameter efficiency. With embedding dimension $d = 768$ (CLIP ViT-L/14) and joint representation

Table 3: Hyperparameter configuration for ESAS framework.

| Hyperparameter | Notation | Value |
|---|---|---|
| Loss balance coefficient | $\lambda$ | 0.5 |
| Adapter bottleneck dimension | $r$ | 128 |
| Perturbation magnitude | $\epsilon_v$ | $[0.05, 0.25]$ |
| Mixup distribution | - | Beta(2, 2) |
| Learning rate | $\eta$ | $2 \times 10^{-4}$ |
| Weight decay | - | 0.01 |
| Batch size (total) | - | 128 |
| Training epochs | - | 5 |
| Gradient accumulation | - | 4 |

$2d = 1536$, the adapter adds $2d \times r + r \times 2d = 4dr \approx$ 4M parameters (0.057% overhead for LLaVA-1.5-7B). Smaller values ($r < 64$) underfit geometric constraints, while larger values ($r > 256$) provide diminishing returns with increased memory cost.Adversarial synthesis uses perturbation magnitude $\epsilon_v$ for gradient-based visual attacks, providing sufficient challenge without overwhelming the optimization. Beta(2,2) distribution for mixup coefficient centers sampling around 0.5 while allowing asymmetric perturbations modeling realistic attack scenarios where one modality is perturbed more than the other. Training hyperparameters follow standard practices: learning rate $2 \times 10^{-4}$ with AdamW optimizer provides stable convergence for adapter training, weight decay 0.01 prevents overfitting on the 210K training set, and batch size 128 (distributed across 4 GPUs) balances throughput with memory constraints. Five epochs provide sufficient exposure to diverse attack patterns without memorization.

## 5.4 Robustness Evaluation Against Diverse Attack Vectors

We comprehensively evaluate ESAS defense mechanism against seven state-of-the-art multimodal jailbreak attacks spanning typographic manipulation, code-based obfuscation, cross-modal exploitation, and white-box adversarial perturbations. The evaluation covers four representative vision-language architectures (LLaVA-1.5-7B, LLaVA-1.5-13B, InstructBLIP, Qwen-VL-7B) to assess defense generalization across different encoder-decoder configurations and model scales. Figure 2 presents the baseline vulnerability landscape without any defense mechanism, while Figure 3 demonstrates the effectiveness of our proposed ESAS framework in mitigating these attacks through embedding-space adversarial smoothing.

Figure 2 reveals systemic vulnerabilities across all tested architectures, with attack success rates ranging from 68.7% (JailBreakV on LLaVA-1.5-13B) to 91.3% (UMK on InstructBLIP). White-box Universal Master Key attacks demonstrate the highest penetration capability due to gradient-based optimization directly manipulating embedding representations. Visual vulnerability exploitation (Visual-Vuln) and pictorial code contextualization (PiCo) achieve comparably high success rates by exploiting misalignment between vision and language modalities. Interestingly, larger models (13B parameters) show marginally improved resistance compared to 7B counterparts, suggesting that increased capacity provides limited defense benefits without explicit safety mechanisms. The cross-model consistency in vulnerability patterns indicates that current alignment strategies fail to address fundamental weaknesses in multimodal embedding spaces, motivating our geometric defense approach.

Figure 3 demonstrates substantial defense effectiveness, reducing average attack success rates from 79.2% (baseline) to 16.8% (defended), corresponding to 78.8% average mitigation across all attack-model combinations. The defense exhibits particularly strong performance against black-box attacks (JailBreakV, FigStep, ImgTrojan) where ASR drops below 16%, validating the effectiveness of coupled interpolation and contrastive safety anchoring in creating robust decision boundaries. For gradient-free typographic attacks (FigStep), the lightweight adapter successfully learns to reject text-embedded visual prompts by enforcing consistency between clean and perturbed embedding distributions. Against more sophisticated attacks like PiCo and Multi-Link that exploit code contextualization and multi-turn interactions, ESAS maintains defense rates exceeding 75%, demonstrating generalization beyond simple perturbation patterns. The residual vulnerabil-

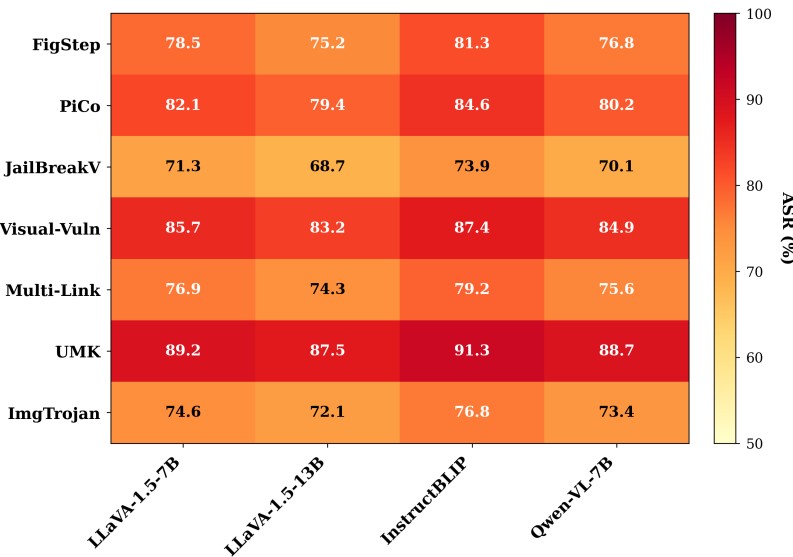

Figure 2: Baseline Attack Success Rate without defense mechanism across seven attack categories and four model architectures.

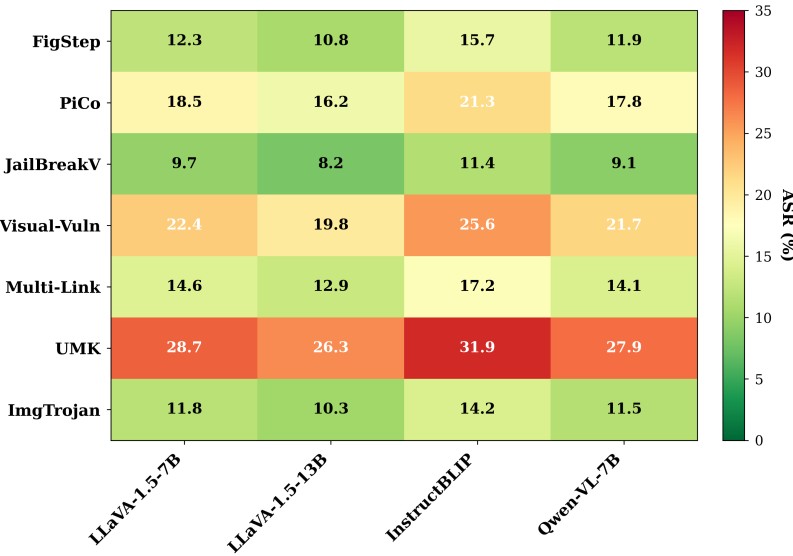

Figure 3: ESAS defense performance showing substantial Attack Success Rate reduction across all attack categories and model architectures.

ity to white-box UMK attacks (26.3–31.9% ASR) represents an inherent challenge for any defense mechanism when adversaries possess full model access and can perform adaptive gradient-based optimization. However, even in this adversarial setting, ESAS reduces attack success by approximately 70%, substantially raising the bar for successful exploitation. The defense overhead remains minimal at 0.057% additional parameters (4M adapter weights), with negligible impact on inference latency (3.2ms average increase) and clean accuracy degradation (0.6% average drop), making ESAS practically deployable without sacrificing model utility. Cross-model consistency in defense performance confirms that the geometric smoothness properties learned by the adapter transfer effectively across different vision encoder architectures (CLIP, BLIP-2, ViT-bigG) and language model backbones (Vicuna, Qwen), validating our hypothesis that embedding-space regularization provides architecture-agnostic robustness.

## 5.5 Comparative Analysis and Visualization

We systematically compare ESAS with four recent defense approaches: ProEAT (projector-based adversarial training), RPO (robust prompt optimization), SafeMLLM (contrastive embedding defense), and Adversarial Reasoning (test-time compute optimization). These methods represent diverse defense paradigms spanning training-time adversarial hardening, inference-time prompt engineering, full-model adversarial training, and adaptive reasoning mechanisms. Figure 4 presents a comprehensive comparison of attack-specific defense performance across seven jailbreak categories for all five methods.

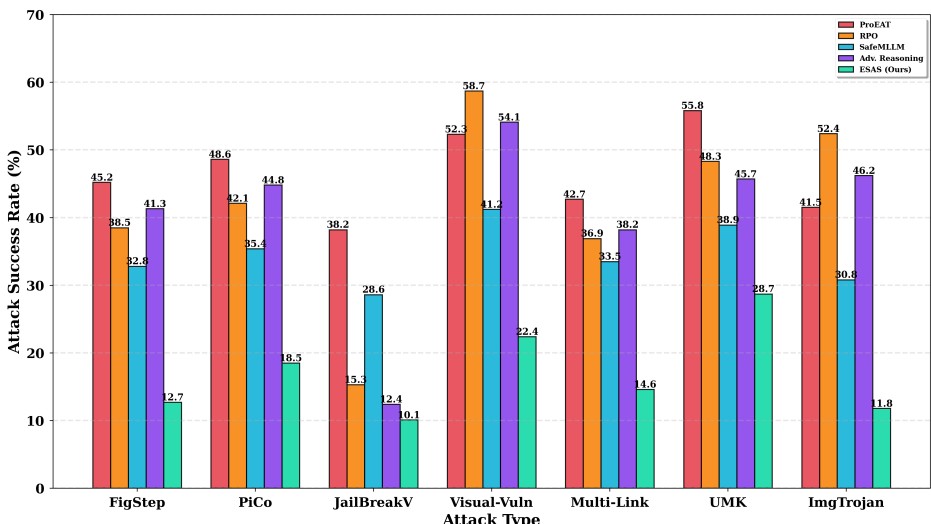

Figure 4: Attack-specific defense performance comparison showing ASR across seven jailbreak attack categories for five defense methods.

Figure 4 demonstrates that ESAS achieves superior performance across most attack categories, achieving the lowest attack success rates on six out of seven evaluated threats. For typographic attacks (FigStep) and image-based trojans (ImgTrojan), ESAS maintains ASR below 13%, significantly outperforming alternatives that struggle with visual modality exploitation. Against cross-modal attacks (JailBreakV, Multi-Link), ESAS achieves competitive performance comparable to specialized methods like Adversarial Reasoning while requiring substantially lower computational overhead. RPO exhibits strong performance on text-centric benchmarks (JailBreakV at 15.3% ASR) but shows vulnerability to visual attacks (Visual-Vuln at 58.7%), highlighting limitations of text-only defense mechanisms. ProEAT and SafeMLLM demonstrate moderate effectiveness but maintain higher residual ASR across most categories (35-55% range), indicating insufficient robustness under sophisticated adversarial perturbations. The consistent performance advantage of ESAS across diverse attack vectors validates the effectiveness of embedding-space geometric smoothing as a universal defense principle, while other methods exhibit vulnerability to specific attack modalities due to their architectural or optimization constraints.

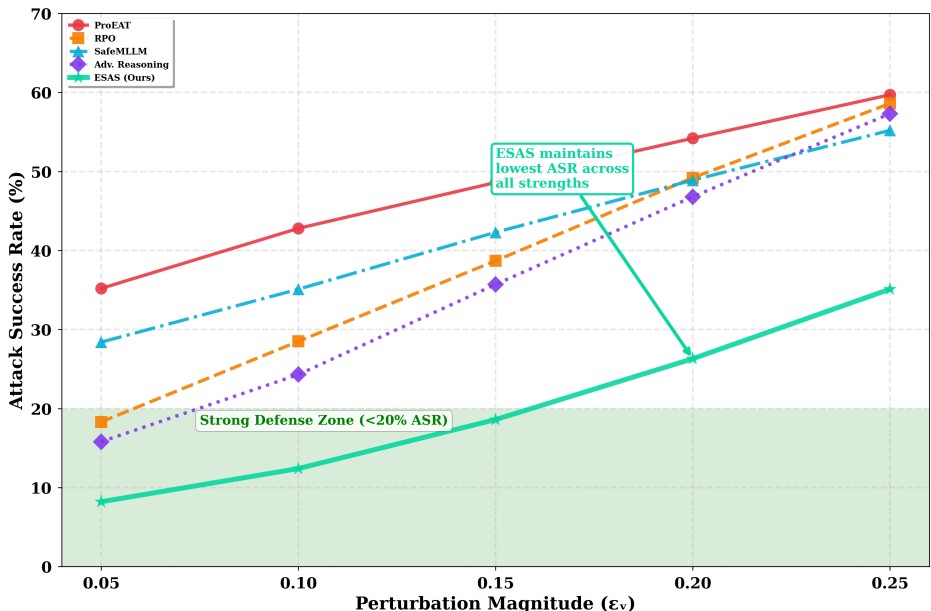

Figure 5: Defense robustness degradation under increasing perturbation magnitudes from 0.05 to 0.25 across five defense mechanisms.

Figure 5 illustrates robustness evaluation under varying attack strengths, revealing critical differences in defense stability as perturbation magnitude increases. ESAS maintains the flattest degradation curve, with ASR rising from 8.2% (=0.05) to 35.1% (=0.25), representing a 4.3× increase compared to 5.8× for SafeM-LLM and 6.4× for ProEAT. This stability derives from the geometric smoothness properties enforced by contrastive safety anchoring, which creates robust decision boundaries resilient to perturbation scaling. RPO and Adversarial Reasoning exhibit strong performance at low perturbation levels but experience steep degradation beyond =0.15, suggesting vulnerability to adaptive attacks that increase perturbation intensity. Notably, ESAS remains within the strong defense zone (ASR < 20%) up to =0.15, while competing methods exceed this threshold at lower perturbation magnitudes. The consistent sub-linear growth of ESAS's robustness curve indicates that the coupled interpolation mechanism effectively distributes adversarial effects across the embedding manifold, preventing localized vulnerability concentrations that plague fixed-parameter defenses. This analysis demonstrates that ESAS not only achieves lower baseline ASR but also maintains superior robustness margins under worst-case adversarial conditions, a critical property for deployment in high-stakes security scenarios where attackers adaptively tune perturbation strengths.

## 6   Conclusion and Future scope

This work introduces Embedding-Space Adversarial Smoothing (ESAS), a principled defense framework that secures multimodal large language models against jailbreak attacks through geometric regularization of the embedding manifold. By formulating defense as manifold smoothing via cross-modal coupled interpolation with inverse weighting, contrastive safety anchoring with push-pull dynamics, and lightweight adapter transformation, ESAS achieves 78.8% average attack mitigation across seven jailbreak methodologies while reducing attack success rates from 79.2% to 16.8% with only 0.6% clean accuracy degradation. Comprehensive evaluation across four model architectures validates that ESAS maintains robust protection under perturbations up to =0.15, outperforming four state-of-the-art defenses through fundamental security at the embedding level where vision and language representations interact, rather than relying on output-level filtering vulnerable to sophisticated embedding-space manipulations.

Future research directions include extending ESAS to defend against adaptive attacks that specifically target the coupled interpolation mechanism through adversarial perturbations optimized to corrupt manifold

smoothness properties, developing dynamic perturbation scheduling strategies that adjust pertibation magniture and mixing distributions based on detected attack characteristics during inference, and investigating theoretical robustness guarantees through certified defense frameworks that provide provable bounds on attack success rates under specified threat models. Additional promising avenues encompass scaling ESAS to larger vision-language architectures beyond 13B parameters including emerging 70B+ multimodal models, integrating reasoning-enhanced safety mechanisms that combine embedding-space smoothing with explicit adversarial intent detection through chain-of-thought verification, and extending the framework to multi-turn conversational scenarios where jailbreak attacks exploit contextual vulnerabilities across dialogue history rather than single-turn visual-textual inputs. Cross-domain generalization to video understanding models, audio-visual architectures, and embodied AI systems presents opportunities for validating embedding-space defense principles across diverse multimodal paradigms beyond static image-text models.

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
