# OpenReview forum: "Defending Multimodal Large Language Models Against Jailbreak Attacks Through Embedding-Space Adversarial Smoothing"
_TMLR — Rejected by TMLR_

### Review · Reviewer_DveW · 2025-11-24

**Summary Of Contributions:**

The paper introduces Embedding-Space Adversarial Smoothing (ESAS), a new defense framework for multimodal large language models that mitigates coordinated jailbreak attacks by operating directly in the joint image–text embedding space. ESAS synthesizes adversarial embeddings through gradient-based visual perturbations and harmful text suffixes, performs cross-modal coupled interpolation with inverse weighting to capture asymmetric attack patterns, and trains a lightweight 4M-parameter adapter to transform embeddings while keeping all encoders and decoders frozen. A contrastive safety anchoring mechanism enforces push–pull geometric regularization, pulling interpolated embeddings toward safe regions while repelling them from adversarial ones. Extensive experiments across seven jailbreak attacks and four MLLM architectures show that ESAS reduces attack success rates from 79.2% to 16.8% with minimal clean-accuracy degradation, outperforming four state-of-the-art defenses. Key strengths include its novel geometric perspective, lightweight and modular design, and broad empirical robustness; key weaknesses include limited adaptive-attack evaluation, incomplete theoretical grounding for manifold smoothing, and insufficient analysis of downstream utility trade-offs.

**Audience:**

Yes

**Audience Explanation:**

Jailbreak robustness in multimodal LLMs is an extremely active and urgent research area. This paper offers:
•	A new modeling perspective (embedding-space geometric smoothing)
•	A light-weight, plug-in defense that is easy to integrate into existing MLLMs
The work is particularly relevant to researchers studying:
•	Adversarial robustness in LLMs and MLLMs
•	Vision–language alignment
•	Model safety and alignment
•	Geometric representation learning
Given the rapid deployment of MLLMs in industry, safety-oriented contributions like this are timely and valuable to the TMLR community.

**Claims And Evidence:**

Yes

**Claims Explanation:**

The paper provides extensive experimental evidence supporting its main claims:
1. The evaluation across seven attack categories and four model families is comprehensive, and figures clearly show consistent ASR reductions.
2.  The adapter is indeed lightweight (≈4M parameters), and clean accuracy is rigorously measured.
3.  ESAS is compared against four state-of-the-art multimodal defenses, with results showing consistent superiority.
However, a few areas could benefit from stronger evidence:
1. No adaptive attack experiments where the attacker targets the embedding level purturbation.
2. Limited utility analysis beyond overall clean accuracy.
3. Lack of ablations isolating each ESAS component (interpolation, anchoring, KL regularization, adapter).

**Requested Changes:**

1.	Add adaptive attack evaluations.
Since ESAS is based on a predictable interpolation mechanism and safety anchoring, attackers could optimize specifically against these elements.
- Adaptive suffixes targeting the anchor
- Gradient-based attacks aware of λ and the adapter
- Without testing adaptive attacks, robustness claims remain partially incomplete.
2.	Provide stronger justification or analysis for “manifold smoothing.”
- Currently, the geometric argument is intuitive. The paper would benefit from at least a simple visualization or quantitative metric demonstrating that the embedding manifold becomes “smoother” after ESAS training.
3.	Add more detailed utility evaluations.
- Report downstream task accuracy on standard MLLM benchmarks (VQA, captioning, visual reasoning).
- Clean accuracy alone is insufficient to judge safety–utility trade-offs.
4.	Add ablation studies.
Important components that should be ablated include:
- Cross-modal inverse mixing
- Beta distribution choice
- Safety anchoring push/pull terms

---

### Review · Reviewer_Z3r7 · 2025-12-02

**Summary Of Contributions:**

The paper proposes ESAS, a multimodal jailbreak defense that trains a small adapter on top of frozen encoders by mixing clean embeddings with gradient-perturbed vision embeddings and text embeddings produced via fixed adversarial suffixes, then optimizing a contrastive objective that pulls mixed embeddings toward clean ones. The method reports lower attack success rates on several benchmark attacks across LLaVA, InstructBLIP, and Qwen-VL. However, the evaluation covers only a narrow set of fixed benchmark threat models and does not test several attacker settings that directly correspond to the defense’s assumptions, including adaptive white-box attackers that optimize through ESAS, joint vision–text gradient attackers, white-box input-space adversaries on images or prompts, transfer attackers across models, or attackers designed to target the safety anchoring. Clean and adversarial results are provided only in aggregated heat-maps. These omissions prevent the experiments from supporting the robustness claims.

**Audience:**

Yes

**Audience Explanation:**

ESAS introduces a small, easily integrated adapter on top of frozen multimodal encoders, and the idea of shaping multimodal embedding geometry through adversarial safety anchoring is a concrete mechanism that some researchers in TMLR's audience would be interested in.

**Broader Impact Concerns:**

The paper introduces a “Universal Master Key” attack described as a white-box gradient-based manipulation of internal multimodal embeddings, but the submission does not discuss the broader implications of formalizing such an attack. The authors do not address whether describing an embedding-level jailbreak method raises security concerns for systems that expose internal model gradients or intermediate representations, and they do not comment on any responsible-disclosure considerations.

Although the attack relies on access to internal embedding gradients and therefore appears impractical in many real deployment settings, the submission does not acknowledge this limitation or examine its implications. The paper should discuss both the restricted feasibility of such an attack and the corresponding broader-impact concerns that arise from publishing a general technique for constructing embedding-level white-box jailbreaks.

**Claims And Evidence:**

No

**Claims Explanation:**

The paper evaluates ESAS only against fixed, non-adaptive benchmark attacks. It does not test adaptive white-box attackers that differentiate through the adapter and interpolation, does not evaluate joint gradient-based perturbations over both modalities, and does not include white-box input-space attacks on images or text prompts. It also does not examine transfer attacks across models or attacks specifically designed to target the safety anchoring mechanism. These missing attacker settings are directly relevant to the proposed defense and are required to assess its robustness. Clean-accuracy results and per-attack/per-model adversarial metrics are provided only in aggregated form, which further limits the evidential strength.

**Requested Changes:**

The paper should evaluate attacker settings aligned with ESAS’s design: adaptive white-box optimization through the defended model, joint vision–text gradient attacks, white-box input-space attacks on images and prompts, transfer attacks across models, and attackers intended to target the safety anchoring. It should also report detailed clean-accuracy results and per-attack/per-model adversarial results, and include ablations for ESAS’s main components. These changes are necessary to support robustness claims.

---

### Review · Reviewer_wqTB · 2025-12-07

**Summary Of Contributions:**

Summary:
- This paper proposes Embedding-Space Adversarial Smoothing (ESAS), an embedding-space smoothing technique using cross-modal coupled interpolation and contrastive safety anchoring, for defending multimodal LLMs against jailbreak attacks. The authors' method reduces the average ASR using only a lightweight adapter.

Strengths:
1. The authors formulate the multimodal jailbreak defense as a manifold-smoothing problem and introduce cross-modal coupled interpolation with inverse weighting.
2. The proposed architecture provides end-to-end protection with frozen encoders.
3. The authors systematically evaluated the method across diverse attack types and multiple model architectures. Furthermore, they validated the stability of the defense mechanism by conducting robustness degradation analysis with respect to perturbation magnitude.

Weak Points:
1. The authors' motivation is not clearly presented in the introduction. The motivation and novelty in the introduction need to be further revised.
2. The evidence for the authors' claims is either unclear or insufficient. In particular, the claims made in the proposed method section lack theoretical justification or supporting evidence.
3. Fig.1 may not be clearly readable or intuitive for readers. It needs to be improved in terms of readability and simplicity to better convey what problem is being addressed and how the authors propose to solve it.

**Additional Comments:**

1. The text size and examples in Figure 1 are not clearly visible, which reduces readability. I recommend that the authors improve the visualization of Figure 1 to enhance clarity.
2. The terms "multi-modal" and "multimodal" are used interchangeably throughout the paper. It would be better to unify the term.

**Audience:**

No

**Audience Explanation:**

Figure 1 conceptually illustrates the overall workflow of ESAS, but it does not provide intuition about how "adversarial embedding synthesis" or "contrastive safety anchoring" actually operate in embedding space. This makes it difficult for readers to intuitively understand the core ideas. The method description in Sec.4 also lacks essential assumptions and theoretical justifications, making it difficult for TMLR readers to gain sufficient understanding.

**Claims And Evidence:**

No

**Claims Explanation:**

1. The authors claim that they model asymmetric patterns where attackers simultaneously exploit vision and text modalities with different intensities. However, there is no empirical analysis demonstrating that the relationship between vision/text perturbation intensities in actual jailbreak attacks follows an inverse relationship. Additionally, regarding the Beta distribution the authors mention, there is no explanation for why this specific distribution was chosen, nor any comparison with other distributions.
2. In Section 4.3, regarding the claim that interpolation is performed on geodesic paths connecting clean and adversarial manifold regions, there is insufficient explanation of the assumptions and conditions required for the interpolation to actually occur on geodesic paths.
3. Regarding Section 4.5, there is no mathematical or experimental verification of whether a "stable equilibrium" is actually formed.

**Requested Changes:**

1. Fig.1 needs to be improved in terms of readability and simplicity to convey better what problem is being addressed and how the authors propose to solve it.
2. In Eq. 8-9, a single mixing coefficient $\lambda$ is used. Does this assume a fixed inverse relationship between the two modalities? It would be helpful to provide a clearer explanation of this design choice.
3. The authors claim that contrastive safety anchoring guides embeddings toward "safe regions." Can the authors provide a clearer explanation of the definition of the safety anchor?
4. The specific definition of $L_{jailbreak}$ used in Eq. (4) is not provided in the main text. It is necessary to explain on how "harmful content generation likelihood" is measured.
5. In Table 1, training is performed on SafetyPrompt while evaluation is conducted on different datasets. Were the baseline methods also trained under the same conditions? The training setup for the comparison methods is not clear.

---

### Decision · Action_Editor_prRE · 2026-01-27

**Recommendation:** Reject

**Audience:**

No

**Audience Explanation:**

Despite the importance of the problem setting, the reliability of the findings cannot be established without more rigorous validation.

**Claims And Evidence:**

No

**Claims Explanation:**

The paper proposes a manifold smoothing-inspired defense against multimodal LLM jailbreaks. However, the submission cannot be accepted in its current form due to significant empirical gaps identified by the reviewers. Specifically, the work lacks evaluations against adaptive attacks, ablation studies, and concrete evidence supporting the manifold smoothing motivation. As the authors did not provide a rebuttal to address these critical concerns, rejection is recommended.